# Trends in National Canadian Guideline Recommendations for the Screening and Diagnosis of Gestational Diabetes Mellitus over the Years: A Scoping Review

**DOI:** 10.3390/ijerph18041454

**Published:** 2021-02-04

**Authors:** Joseph Mussa, Sara Meltzer, Rachel Bond, Natasha Garfield, Kaberi Dasgupta

**Affiliations:** 1Department of Medicine, McGill University, Montreal, QC H4A 3J1, Canada; joseph.mussa@mail.mcgill.ca (J.M.); sara.meltzer@mcgill.ca (S.M.); rachel.bond@mcgill.ca (R.B.); natasha.garfield@mcgill.ca (N.G.); 2Centre for Outcomes Research and Evaluation of the RI-MUHC, 5252 boul de Maisonneuve Ouest, Office 3E.09, Montreal, QC H4A 3S5, Canada; 3Department of Obstetrics and Gynecology, McGill University, Montreal, QC H4A 3J1, Canada

**Keywords:** clinical practice guidelines, gestational diabetes mellitus, pregnancy, diabetes mellitus, neonatal complications, national, screening, diagnosis, one step, two step, prevalence

## Abstract

Canada’s largest national obstetric and diabetology organizations have recommended various algorithms for the screening of gestational diabetes mellitus (GDM) over the years. Though uniformity across recommendations from clinical practice guidelines (CPGs) is desirable, historically, national guidelines from Diabetes Canada (DC) and the Society of Obstetricians and Gynaecologists of Canada (SOGC) have differed. Lack of consensus has led to variation in screening approaches, rendering precise ascertainment of GDM prevalence challenging. To highlight the reason and level of disparity in Canada, we conducted a scoping review of CPGs released by DC and the SOGC over the last thirty years and distributed a survey on screening practices among Canadian physicians. Earlier CPGs were based on expert opinion, leading to different recommendations from these organizations. However, as a result of the Hyperglycemia and Adverse Pregnancy Outcome (HAPO) study, disparities between DC and the SOGC no longer exist and many Canadian physicians have adopted their recent recommendations. Given that Canadian guidelines now recommend two different screening programs (one step vs. two step), lack of consensus on a single diagnostic threshold continues to exist, resulting in differing estimates of GDM prevalence. Our scoping review highlights these disparities and provides a step forward towards reaching a consensus on one unified threshold.

## 1. Introduction

In Canada, gestational diabetes mellitus (GDM) is the most frequent endocrinopathy of pregnancy [1]. It is defined as glucose intolerance resulting in hyperglycemia with first recognition or new onset during pregnancy, but the specific glycemic thresholds for its diagnosis are a persistent subject of debate. Notwithstanding differences in definitions and their application over the last three decades, the prevalence of GDM is rising around the world [2]. Increases in obesity rates, maternal age, and ethnic diversity and changes in diagnostic thresholds have likely contributed to this shift. 

In Canada, as in much of the world, there has been debate concerning: (a) the appropriate timing and method for screening, specifically the utility of a 50 g glucose challenge test (GCT) prior to an oral glucose tolerance test (OGTT) with a higher glucose load (one step vs. two step approach); (b) what constitutes the most appropriate glucose load (e.g., 75 g vs. 100 g in glucose tolerance testing); (c) the specific glucose threshold values above which a test is considered abnormal at different time points following the glucose load; and (d) the number of abnormal values required to warrant a GDM diagnosis [3]. Although the hyperglycemia observed in GDM typically resolves post-partum, GDM history is a risk factor for incident diabetes mellitus [4], hypertension [5], and cardiovascular disease later in life [6]. The original definitions of GDM were conceived with a focus on the future risk of maternal diabetes mellitus [7]. However, GDM is associated with other short-term and long-term health outcomes in both the mother and her offspring that are now considered in selecting diagnostic thresholds [3,8].

Since the initiation of the 2008 Hyperglycemia and Adverse Pregnancy Outcome (HAPO) study [8], there have been a growing number of epidemiological analyses based on HAPO data and other data sources demonstrating compelling evidence of associations between GDM and a wide array of adverse neonatal complications [9,10,11,12,13]. In the shorter term, several analyses have demonstrated that maternal glucose intolerance may increase risk of pre-term delivery, perinatal morbidity and mortality, neonatal hypoglycemia, macrosomia, neonatal hyperinsulinemia, and congenital malformations [8,9,10]. In the longer term, GDM is also associated with offspring complications such as childhood obesity, dyslipidemia, and future diabetes mellitus later in life [11,12,13]. 

Given the consequences that GDM may have on both the health of the mother and her offspring, it is important to detect its presence in pregnancy as early as possible. Though uniformity across recommendations from Canadian clinical practice guidelines (CPGs) is desirable and would be less confusing for practitioners, historically, national guidelines from two key organizations, Diabetes Canada (DC) and the Society of Obstetricians and Gynaecologists of Canada (SOGC) have differed. In this scoping review, we discuss: (1) the evolution of national recommendations for the screening of GDM in Canada over the last thirty years by both DC (formerly known as the Canadian Diabetes Association) and the SOGC; (2) the degree of variability in screening practices adopted by Canadian health care providers in their practice; and (3) the impact of varying diagnostic criteria on the estimates of GDM prevalence in Canada. 

## 2. Study Design and Methods 

We conducted a scoping review of CPGs from DC and the SOGC and a voluntary, online survey of health care providers dedicated to GDM care.

### 2.1. Search Strategy

Published literature was retrieved through searches in five electronic bibliographic databases (The Cochrane Library, PubMed, CINAHL, Web of Science and SCOPUS) from 1 January 1964 to 30 November 2020. Subject headings and key MeSH terms included “national recommendations”, “clinical practice guidelines”, “diabetes mellitus”, “pregnancy”, “gestational diabetes mellitus”, “screening”, “diagnosis”, “one step” and “two step”. The search strategy was based on three key concepts: (1) pregnancy (study population); (2) GDM (exposure); and (3) screening and diagnostic parameters (outcome). Restrictions for language (limited to English and/or French materials) and geographic location (Canada; limited to national-level recommendations) were applied. In addition, the reference lists of all identified CPGs were examined to identify other Canadian national guidelines not captured in our search. The electronic search and the eligibility of the guidelines were independently assessed by two reviewers (JM, KD) and discrepancies were resolved through discussion. 

In addition, several interviews were conducted with one of the co-authors (SM) to discuss the history of GDM screening and aid in the identification of key Canadian guidelines over the years. SM served as the Steering committee co-chair in the development of the 1998 DC CPG for the management of diabetes in Canada; she holds extensive, substantive knowledge on the diagnostic criteria recommended by Canadian CPGs over the years.

### 2.2. CPG Selection and Data Extraction

CPG recommendations were retained if they met the following criteria: (1) CPGs included recommendation for screening, diagnosing, and managing diabetes mellitus during pregnancy; (2) recommendations were made at the national level (CPGs specific to a local region of Canada were excluded). Abstracts, case reports, study protocols, commentaries, observational studies, reviews, randomized controlled trials, and meta-analyses were excluded. The full-text articles of all relevant guidelines were reviewed (JM).

Data extraction captured the following information from CPGs retained: (1) year of publication; (2) recommended population for GDM screening; (3) method/test for screening and diagnosis; (4) number of abnormal values required for diagnosis; (5) glucose thresholds to warrant a GDM diagnosis after initial screening test and/or diagnostic testing (fasting glucose, 1 h after loading, 2 h after loading, 3 h after glucose load); (6) estimated prevalence of GDM. One author (JM) extracted data from all eligible CPGs which underwent review by another (KD). Discrepancies were resolved through discussion. 

### 2.3. Survey Distribution

We also conducted a survey among physicians from the Canadian Diabetes in Pregnancy (CanDIPS) study group to determine what GDM screening practices they are currently using in clinical practice (Figure A1). The survey link was distributed to CanDIPS members via electronic mail by one of the co-authors (RB). 

## 3. Results

### 3.1. Search Results

The initial search identified 38 CPGs. A total of nine CPGs were screened for eligibility after removal of duplicates (*n* = 6) and local CPGs specific to a region in Canada (*n* = 23). In total, nine national CPGs were retained (Figure 1).

### 3.2. CPG Characteristics

National guidelines were published by the SOGC [14,15,16,17], the largest national obstetrical society, and DC [18,19,20,21,22], the largest national society of diabetology. Since the release of the first Canadian CPGs to address diabetes during pregnancy by the SOGC in 1992 [14], this organization released subsequent, updated versions of its guidelines in 2002 [15], 2016 [16] and 2019 [17]. DC released five national guidelines on screening, diagnosing and managing GDM in Canada; these include the first release in 1998 [18] followed by revised guidelines published in 2003 [20], 2008 [20], 2013 [21] and 2018 [22]. 

Several key differences in recommendations regarding the necessity and benefits of universal screening, the appropriate method for GDM screening, and appropriate glucose cut-off thresholds exist between national guidelines published from each of these societies. 

### 3.3. The Origin of Defining GDM

The increased risk of obstetrical complications associated with GDM was first detailed in an issue of *Diabetes* authored by Dr. J.P. Hoet in 1954 [23]. Shortly after the release of this publication, the National Institutes of Health (US) initiated a program focused on the epidemiology of chronic disease, a program joined by Dr. John O’Sullivan in the late 1950s [24]. During the era following World War II, there was widespread interest and controversy around the globe regarding the method of diagnosing GDM among pregnant women. At this time, Canadian physicians relied on “elevated” glucose values following a 100 g OGTT to warrant a diagnosis of GDM; thresholds were defined vaguely and left to the interpretation of the individual physician. 

To generate evidence, Dr. O’Sullivan conducted a prospective cohort study (New York, NY, USA) [7]. He challenged 752 pregnant women in their second or third trimester (“pregnancy cohort”) with 100 g oral glucose loads and measured whole blood glucose levels, at baseline, 1 h, 2 h, and 3 h after the load, using the Nelson–Somogyi method and rounding to the nearest whole number. He calculated the means and standard deviations (SD) at each of these time points, considering two SD above the mean to be elevated, such that 5% of the pregnancy cohort would be considered abnormal. Applying only one SD and corresponding glucose thresholds would have resulted in a higher proportion of women to have been considered to have GDM [25]. O’Sullivan believed that this would lead to psychologic ill effects (i.e., depression, anxiety, eating disorders) and unnecessary long-term follow up of patients with only mild glucose intolerance [7]. These concerns were expected to pose significant increases in economic burden, while only offering minimal benefit towards preventing maternal diabetes mellitus later in life. Similar concerns are part of today’s debates concerning optimal screening methods.

Subsequently, O’Sullivan and statistician, Dr. Mahan, defined GDM as two or more elevated values of glucose among the four time points. This definition was published as the first set of statistically-based criteria to define glucose intolerance during pregnancy in 1964 (fasting, 5.0 mmol/L; 1 h, 9.2 mmol/L; 2 h, 8.1 mmol/L; 3 h, 6.9 mmol/L) [7]. O’Sullivan conducted several follow-up studies during the 1960s and re-applied his pre-defined thresholds of “elevated glucose” to define GDM among a different group of 1013 women tested during pregnancy. Women were followed for 5–10 years post-partum and results indicated that 22% of women with GDM in the cohort later developed diabetes mellitus within 7–8 years after their pregnancy [26]. These findings were consistent with several holding theories at the time explaining that GDM may be associated with post-partum maternal diabetes mellitus; shortly after publication, his criteria were accepted on the basis of risk assessment for future maternal diabetes mellitus [18,26,27].

### 3.4. Evolution in Screening Approaches: Early Adoption of the 50 g GCT

Some physicians in Canada had slowly begun to adopt thresholds proposed by O’Sullivan due to their increasing recognition in the late 1960s to early 1970s [12,24]. Individual physicians used their own discretion to decide who required a 100 g OGTT. At this time, the physician’s decision was based on the presence of known risk factors for GDM during this period, which were predominantly limited to renal glycosuria during pregnancy, previous history of large infants at birth, and family history of diabetes mellitus [25]. However, in the pregnancy cohort followed by O’Sullivan, restricting screening to those defined as “at risk” by these risk factors demonstrated insufficient sensitivity (63%) and specificity (57%) for the detection of GDM [28]; 37–50% of women with GDM would remain undiagnosed [28,29]. In 1973, O’Sullivan and Mahan recommended the use of a screening test in all pregnant women, the 50 g 1 h glucose challenge test (GCT), to improve the detection of women with GDM without the need to subject all of these women to a longer 100 g tolerance test [28]. Using the Nelson–Somogyi method, a threshold of 7.2 mmol/L at one hour post-ingestion of the 50 g glucose load was 79% sensitive and 87% specific for GDM in his pregnancy cohort [7]. Although O’Sullivan demonstrated the positive predictive value (PPV) of the 50 g GCT to be merely 14%, the negative predictive value (NPV) was 99.4%; these results indicate that 50 g GCT screening tests produced an excess of false positives but minimal false negative results [12]. Since the pregnancy cohort underwent both the 50 g GCT screening test followed by a 100 g OGTT, O’Sullivan’s proposed method allowed for strong GDM case ascertainment, which quickly became adopted as the gold standard.

### 3.5. Evolution of O’Sullivan’s Proposed Criteria

In the late 1970s, the US National Diabetes Data Group (NDDG) endorsed O’Sullivan’s criteria with several slight modifications, but determined that plasma glucose should be used instead of whole blood values [30]; therefore, they increased the diagnostic thresholds (to FPG 5.8 mmol/L; 1 h 10.6 mmol/L; 2 h 9.2 mmol/L; 3 h 8.0 mmol/L) given that the glucose content present in whole blood is less than that found in plasma (Table 1). With endorsement from the NDDG, widespread screening for GDM using these modified criteria grew rapidly across the globe, including application in clinical practice among many Canadian physicians during the mid 1980s. In 1982, Drs. Carpenter and Coustan proposed replacing the Nelson–Somogyi method with more accurate enzyme-based assays [31]. The Nelson–Somogyi method measures all reducing substances present in whole blood and is not specific for glucose; this typically results in glucose measurements 11–15% higher than more specific enzyme-based assays [25]. With these assays, Drs. Carpenter and Coustan lowered the diagnostic cut off for GDM relative to values proposed by the NDDG (Table 1). In Canada during the 1980s, physicians variously implemented the O’Sullivan, NDDG, and Carpenter–Coustan criteria. These reference thresholds were an improvement over the more subjective approaches to GDM diagnosis that had previously been used, yet there remained a wide variation in clinical practice.

### 3.6. Universal vs. Selective Screening 

The 1992 SOGC CPG [14] recommended universal screening at 24 to 28 weeks with the 50 g GCT and progression to a 100 g OGTT if glucose values met or exceeded 7.8 mmol/L (1 h post-glucose ingestion). In fact, 84% of Canadian physicians at this time had adopted this approach even prior to the 1992 SOGC guidelines, given the validation of the 50 g GCT screening test (improved sensitivity and specificity) by O’Sullivan twenty years prior [15]. Several years later, the second Canada-wide CPG to encompass diabetes mellitus in pregnancy was published by DC in 1998 [18]. Emerging evidence at this time suggested that women at low risk could be exempt from screening [32]. Selective screening was endorsed in the 1998 DC CPGs and subsequently adopted by the 2002 SOGC guidelines. Advantages of selective screening were reductions in the burden of screening on pregnant women and the health care system. Low-risk individuals were defined as those 25 years of age or younger, pre-pregnancy BMI <27 kg/m^2^ (the SOGC) or “non-obese” (DC), Caucasian ethnicity or other ethnic group with low diabetes mellitus prevalence, no previous history of GDM or glucose intolerance, no history of GDM-associated adverse pregnancy outcomes (i.e., macrosomia) and no family history of diabetes mellitus in first-degree relatives [15,18]. Despite these recommendations, many physicians in Canada still chose to practice universal screening since the majority of pregnant women do not meet all criteria needed to be considered low-risk [12,33]. In 2003, DC revised their national guidelines to recommend universal screening [19]. Since the release of their 2003 CPG, DC have consistently advocated for universal screening in their 2008, 2013 and 2018 CPGs because the expert panel holds that: (a) selective screening allows for undiagnosed cases of GDM among women who do not have risk factors [34]; (b) most Canadian women (90%) do not meet the criteria to be considered low risk, rendering selective screening complicated and unnecessary (supported by evidence from a cohort of 1655 pregnant women in Australia [35]); and (c) although more expensive in the short-term, universal screening for GDM may reduce the long-term costs and burden of future complications in the mother and offspring [18,19,20,21,36].

After its 2002 CPGs, the SOGC did not provide an update until 2016. The 2002 guidelines had recommended selective screening and the physicians’ choice between a 75 and 100 g OGTT. In 2016, the SOGC aligned with the 2013 DC CPGs, recommending that all pregnant women be screened at 24 to 28 weeks’ gestation with a 75 g OGTT [16].

### 3.7. Diagnostic Approaches: Variations in the Testing Times and Recommended Glucose Loads to Be Administered for OGTT

The 1992 SOGC guidelines recommended the 50 g GCT followed by a 100 g 3 h OGTT with at least two abnormal values to warrant a diagnosis of GDM. During this time, the application of a 50 g GCT (screening test) followed by a 100 g 3 h OGTT (diagnostic test) was commonly practiced in most countries [12]. The 1998 DC guidelines advocated the 75 g 2 h OGTT with at least two abnormal values as the preferred diagnostic method following a 50 g GCT screening test. The recommendation for a 75 g OGTT was based on the fact that: (a) non-pregnant criteria for diabetes mellitus were based on a standardized 75 g OGTT and (b) the test allows for less blood sampling, less time for testing, lower costs, and less nausea as a result of the lower glucose load administered [18]. However, they retained the 100 g 3 h OGTT as an alternative option given its widespread application in North America but with specification of Carpenter–Coustan thresholds. Carpenter–Coustan thresholds are more inclusive with lower values of 5.3, 10.0, 8.6 and 7.8 mmol/L (Table 1).

Similarly, in 2002, the SOGC adopted the 75 g 2 h OGTT as a diagnostic tool with at least two abnormal values [15], in addition to the 100 g OGTT that its previous guidelines had endorsed [14]. The adoption of the 75 g OGTT approach was consistent with recommendations from the 1998 DC, 1998 American Diabetes Association, 1999 World Health Organization and 2001 American Congress of Obstetricians and Gynecologists guidelines available at the time. Both options were included due to an “absence of clear, comparative trials” [15]. The 2002 SOGC guidelines applied Carpenter–Coustan criteria to the 75 g OGTT while recommending both NDDG or Carpenter–Coustan thresholds for the 100 g OGTT (Table 1), the latter test having higher test sensitivity for GDM relative to the 75 g OGTT. Eventually, the 100 g 3 h OGTT alternative was removed in the revised 2003 DC guidelines due to the inconvenience, poor tolerance and costs associated with the three hour test [19]. DC guidelines continued to require all women to be screened via 50 g GCT and at least two abnormal values of plasma glucose during an OGTT to identify GDM in their 2003 guidelines. Similarly, upon the SOGC’s recent updates in 2016 and 2019, their guidelines also have removed recommendations for the 100 g OGTT and endorse that Canadian providers apply the 75 g OGTT for diagnostic purposes [16,17].

### 3.8. Variation in Screening and Diagnostic Approaches: Debates on Glucose Thresholds Prior to Efforts for International Consensus in 2008

All of the 1992/2002 SOGC and 1998/2003/2008 DC recommendations were based on substantive expert opinion due to a scarcity of high-quality evidence at this time [33]. The early versions of the SOGC (1992, 2002) and DC (1992, 2003, 2008) share consensus on several criteria including: a) applying the 50 g GCT screening technique; b) the requirement of plasma glucose levels > 7.8 mmol/L (at 1 h post-ingestion) following a 50 g GCT to allow for progression towards an OGTT; c) the requirement of plasma glucose levels >10.3 mmol/L (at 1 h post-ingestion) following a 50 g GCT to warrant an immediate diagnosis of overt diabetes mellitus; and d) two abnormal OGTT values to conclude a diagnosis. However, over the years, there has been uncertainty about the specific levels of plasma glucose required to prevent complications in the mothers and offspring. Therefore, the cut-off thresholds warranting a diagnosis of GDM following a 100 g and 75 g OGTT have typically differed across these organizations over the years. 

In terms of diagnostic approaches using the 100 g OGTT approach, guidelines from the 1992/2002 SOGC differ slightly from those proposed by the 1998 DC guideline. The early SOGC guidelines [14,15] suggested application of both Carpenter–Coustan and NDDG criteria when conducting the 100 g OGTT (Table 1) and were based on earlier guidelines from American Congress of Obstetricians and Gynecologists. They suggested that physicians consider either threshold, given insufficient evidence demonstrating clear benefit of one set of criteria over another. In contrast, the 1998 DC guideline [18] recommended only Carpenter–Coustan criteria be applied to define glucose thresholds following administration of the 100 g OGTT (alternative approach), as also recommended by the 1998 American Diabetes Association. Application of Carpenter–Coustan criteria generally leads to increased test sensitivity, given that the thresholds are lower relative to NDDG criteria and thus more inclusive.

Although both the 2002 SOGC and 1998/2003/2008 DC guidelines allowed for diagnostic testing using the 75 g OGTT, cut-off thresholds using this approach differed across guidelines published from these two organizations. The 2002 SOGC’s recommendations for 75 g OGTT thresholds [15] are based primarily on Carpenter–Coustan criteria as applied to the 100 g OGTT with no inclusion of upper NDDG criteria, given that women are administered a smaller glucose load relative to the 100 g OGTT (Table 1). Meanwhile, the guidelines from the 1998/2003/2008 DC guidelines had suggested higher thresholds relative to the lower thresholds from the 2002 SOGC guidelines when testing with a 75 g OGTT (Table 1). The DC expert panel argue that the previous Carpenter–Coustan and NDDG criteria are based on O’Sullivan’s original data from the pregnancy cohort; the mean fasting levels of glucose found in two prospective, multicentre studies (~4000 pregnant women) were slightly different [37,38]. The derivation of 2 SD above the mean plasma glucose in these cohort of women leads to thresholds that lie between the Carpenter–Coustan and NDDG criteria, as suggested in their proposed thresholds.

### 3.9. The HAPO Study and Application of Its Results by the International Association of Diabetes and Pregnancy Study Groups (IADPSG)

Although estimates of GDM prevalence can be derived from health administrative database definitions for GDM that rely on physician billing and hospitalization diagnostic codes, the widespread variations in screening approaches result in varying definitions of GDM over the years (based on available guidelines at this time) and across physicians (Table 1). The 2008 HAPO study [8] was conducted in response to the persistent need for a standardized, internationally-agreed-upon GDM diagnostic criteria that took into account both maternal and offspring outcomes. The original investigation was a multicentre, five-year, prospective cohort study. The investigators recruited more than 25,000 pregnant women in nine countries between July 2000 and April 2006 willing to undergo a 75 g OGTT between 24 and 32 weeks of gestation. Participants were ethnically diverse, represented by 48% Caucasians, 29% Asians, 12% Blacks and 8% Hispanics [8]. The four primary outcomes included cesarean delivery, clinical neonatal hypoglycemia (as noted in medical records), LGA status (defined as birth weight > 90th percentile for gestational age, gender, ethnicity, parity) and hyperinsulinemia (cord serum C-peptide >90th percentile for the study group as a whole). Secondary outcomes included pre-term birth, shoulder dystocia, pre-eclampsia, admission for neonatal intensive care, percent body fat and hyperbilirubinemia.

For categorical analyses, fasting plasma glucose (FPG) levels were classified a priori into seven different categories each in 0.2775 mmol/L increments representing the SD of that value. A similar method was applied to categorize plasma glucose septiles corresponding to 1 and 2 h post-75 g glucose loading. These analyses demonstrated that the association between categorized maternal glucose and frequency of each of the primary outcomes was linear and continuous across time points. The HAPO investigators did not conclude any specific recommendations, given that their analyses demonstrated no clear threshold at which to define GDM, further fuelling controversies around appropriate cut-off points to guide systems of care. Subsequently, a meeting was convened in Pasadena under the umbrella of the IADPSG to develop a consensus regarding the appropriate diagnostic criteria, given findings from the HAPO study. During the workshop conference in 2008, the IADPSG panel agreed that several of the adverse outcomes initially studied were not equally important for devising diagnostic criteria; the panel concluded that hyperinsulinemia based on C-peptide, neonatal body fat and LGA outcomes should comprise the basis for determining diagnostic thresholds, considered as one composite primary outcome [9].

At each time point, individuals with blood glucose within the third septile (representing the mean) were chosen as the reference and compared to those with mean glucose higher by 1 SD (0.38 mmol/L for FPG, 1.71 mmol/L for 1 h PG, 1.30 mmol/L for 2 h PG) to produce odds ratios (OR) for the composite outcome developed by the IADPSG. The IADPSG considered ORs of 1.5, 1.75 and 2.0, and ultimately focused on an OR of 1.75, defining diagnostic thresholds (fasting glucose: 5.1 mmol/L, 1 h glucose: 10.0 mmol/L, 2 h glucose: 8.5 mmol/L) in terms of correspondence to 75% increased odds (OR = 1.75) of cord serum C-peptide > 90th percentile, neonatal body fat > 90th percentile, and LGA at each time point. Setting thresholds based on an OR = 1.5 was believed to lead to a diagnostic test with low PPV (generating an excess of false positives) with 20% being diagnosed with GDM [9]. Of note, the 2 h glucose threshold corresponding to OR = 1.5 was 7.8 mmol/L which was also the 2 h glucose threshold used to diagnose GDM in other earlier guidelines (i.e., 1999 World Health Organization). Glucose thresholds corresponding to an OR = 2.0 were believed by the IADPSG to lack sensitivity. Thresholds corresponding to an OR = 1.75 identified 16.1% incidence in the HAPO cohort [9].

In addition to the glucose thresholds, the IADPSG investigators also decided that only one abnormal OGTT value should be required to conclude a diagnosis of GDM, given that the corresponding glucose thresholds were modelled independently across each time point. They further recommended directly conducting a 75 g OGTT without the necessity for a 50 g GCT screening test (one-step approach) since women in the cohort did not undergo 50 g GCT screening and glucose thresholds corresponding to 75% increased odds of the primary outcome were modelled solely considering OGTT values. In addition, a one-step test was endorsed as the preferred method by the IADPSG due to the ease of administrating the test, given that a woman may not always return to the clinic for an OGTT following screening. The IADPSG task force has also endorsed universal screening and recommended that a fasting plasma glucose > 7 mmol/L or HbA1c > 6.5% discovered in the early stages of pregnancy (before 24 weeks) should be identified as pre-existing diabetes mellitus. These recommendations are published in the 2010 IADPSG guidelines [9].

### 3.10. Uniform CPG Recommendations: Recent Trends in Glucose Thresholds and Updated CPGs in Response to the 2008 HAPO Trial and the 2010 IADPSG Guidelines

The plasma glucose cut off suggested in the 2013 DC recommendations were the first Canadian guidelines to adopt the findings from the IADPSG expert panel [21]. These new guidelines introduced the notion of two different but acceptable approaches to identifying GDM: (a)A two-step approach (preferred by DC) which involves screening (50 g GCT) and diagnostic testing (75 g OGTT) similar to previous guidelines but basing thresholds on HAPO values signaling an OR of 2.0, rather than 1.75 as adopted by the IADPSG [9]. The higher OR corresponds to less inclusive glucose thresholds, aimed to somewhat offset increases in workload, patient burden (glucose monitoring) and associated costs [21].(b)A one-step approach (alternative approach) as endorsed by the IADSPG and using the IADSPG thresholds based on the OR of 1.75 as discussed previously. The IADPSG has endorsed one-step testing as the only approach to diagnosing GDM and have concerns that many women are unable to return following a 50 g GCT. Ancillary data, along with previous retrospective studies [39], have demonstrated that most women (82%) return to complete a 75 g OGTT following a screening test and that this is not a major concern in Canada.

Recommendations from earlier versions of DC guidelines (1998/2003/2008) also suggested that plasma glucose levels > 10.3 mmol/L following a 50 g GCT were sufficient to conclude a diagnosis of GDM. Currently, no high-quality evidence exists to endorse a specific glucose threshold at which the 50 g GCT can be used for diagnostic purposes. Although Carpenter–Coustan’s original work in the 1980s demonstrated that a threshold of 10.1 mmol/L had a PPV of 95% [31], recent evidence has demonstrated equivocal findings. For example, in a retrospective cohort study of 14,771 women screened for GDM between 1988 and 2001, a 50 g GCT threshold of 11.1 mmol/L only demonstrated 84% PPV while >12.8 mmol/L demonstrated 100% PPV [40]. Furthermore, pregnant women with GCT values >11.1 mmol/L in the cohort were more than twice as likely to have caesarean delivery than women below this cut off (OR = 2.24, 95% CI 1.19–4.21). Given these findings, the 2013 DC expert panel agreed that increasing this threshold to >11.1 mmol/L was warranted in order to avoid additional testing for women with markedly elevated levels of glucose and to minimize delays to treatment [21,33]. While a higher glucose threshold increases specificity (lowering the risk of a false-positive results), the trade-off is reduced sensitivity which allows women with severe hyperglycemia to remain untreated for some period of time (until administered a diagnostic test).

While DC has consistently updated its GDM recommendations over the years, the SOGC provided its most recent updates in 2016 and 2019, more than a decade after its last release in 2002. The 2016/2019 SOGC CPGs have now reached a consensus with the 2013/2018 DC Canada CPGs, proposing similar methods of screening and diagnosis with the release of DC’s latest guidelines. This includes the recommendation of universal screening, abandoning the 100 g OGTT, shifting the values required for an immediate diagnosis following a 50 g GCT to higher thresholds (>11.1), adopting the one-step and two-step approach with DC-endorsed cut-off thresholds, and identifying new risk factors for GDM that warrant earlier screening (i.e., polycystic ovarian syndrome, corticosteroid use) since its last update [16,17]. While the 2013/2018 DC expert panel classifies this recommendation for early screening among women with multiple clinical risk factors as based on expert consensus opinion [21,22], the 2016/2019 SOGC panel considers this recommendation to be based on evidence from well-designed cohort studies [16,17].

### 3.11. The Impact of Changing Diagnostic Criteria on Prevalence across Canada

The current prevalence of GDM in Canada has seen a drastic rise, with quadruple the number of women diagnosed with GDM over the last two decades (Table 1). Apart from increases in obesity rates, maternal age and ethnic diversity, changes to the diagnostic criteria for GDM over the years have largely contributed to the observed rise in GDM prevalence [41]. Findings from a large, population-based study (1,109,605 women delivering between 1996 and 2010 in Ontario), conducted by Feig et al. [42], revealed that the age-adjusted incidence rates of both GDM (2.7% to 5.6%, *p* < 0.001) and pre-GDM (0.7% to 1.5%, *p* < 0.001) doubled from 1996 to 2010. Since the Canada-wide adoption of the 2010 IADPSG CPG recommendations for one-step testing, first initiated in the 2013 DC and 2016 SOGC CPGs, the national prevalence of GDM has shifted from approximately 3.7–6.5% to now 7–16% (Table 1). Traditionally in Canada, GDM was diagnosed using the two-step approach; however, following the release of guidelines from the IADPSG in 2010, the current criteria now recommends both two-step testing (preferred approach) and one-step testing (alternative approach). 

As previously mentioned, DC had re-calculated their 2013 thresholds [21] for the two-step approach to correspond with an OR = 2.0 from the HAPO study [8], leading to thresholds similar to those proposed since their 2003 guidelines. However, the prevalence of GDM in the Canadian population ascertained through these two guidelines will differ due to changes in sensitivity from the revised criteria for testing. The reason for this disparity stems from another modification implemented in their 2013 guidelines: only one abnormal value during the post-load time is required to determine a diagnosis of GDM (as opposed to two abnormal values required previously), thus increasing the test sensitivity of these new diagnostic criteria [41]. Furthermore, an increase in the nationwide prevalence of GDM over the last decade can be attributed to the updated Canadian guidelines now recommending one-step testing as an alternative approach with lower thresholds that are more diagnostically sensitive for GDM (5.1, 10.0, and 8.5 mmol/L; Table 1).

In a previous prospective cohort study of 2500 pregnant women, conducted by Agarwal et al. [43], the investigators aimed to compare the differences between several international expert panel diagnostic criteria for GDM and the implications of switching to the one-step approach as endorsed by the IADPSG. Agarwal et al. demonstrated that switching from DC’s two-step preferred approach to the one-step approach led to a 15.3% increase in the prevalence of GDM among the study group. In comparison to the 2003 DC CPGs, applying the IADPSG’s one-step approach led to a 36.1% increase in the prevalence of GDM among the women. Similarly, in another retrospective study conducted in Ontario by Pouliot et al. [44], they found switching from two-step to one-step testing increased the prevalence of GDM from 10.8% to 17.6% among the study cohort. This substantial variability in screening practices adds to the complexity of calculating the true prevalence of GDM in Canada.

### 3.12. The Impact of Changing Diagnostic Criteria on Health Care Economic Costs

In terms of the impact on resources within the Canadian context, application of the one-step approach is believed to decrease the laboratory workload, yet pose more immediate costs to the patient and health care system [45]. In another cost minimization analysis [46], Meltzer et al. compared the cost implications of switching from the two-step approach to the one-step approach among a subset of 1500 pregnant, Canadian women attending tertiary care (Royal Victoria Hospital, Montreal, Quebec). Women who presented for GDM screening and consented to participation in the study were randomized to Group 1 (1 h, 50 g GCT + 3 h, 100 g OGTT with 2002 SOGC NDDG criteria), Group 2 (1 h, 50 g GCT + 2 h, 75 g OGTT with 2013/2018 DC criteria) and Group 3 (2 h, 75 g OGTT alone with 2013/2018 DC criteria for the 1 step approach). Meltzer et al. demonstrated that the two-step approach, using either a 75 or 100 g OGTT, was found to be less costly with similar diagnostic sensitivity to the one-step approach. While GDM prevalence was found to be similar across all three groups (3.7%, 3.7% and 3.6%, respectively), the total costs per woman screened were as follows: Group 1, $91.61 CAN; Group 2, $89.03 CAN; Group 3, $108.3 CAN. Total costs included direct medical costs, direct transportation costs and indirect time costs. The higher total costs of one-step testing were attributed to increased medical costs (blood draws and laboratory analysis) and the indirect time costs, which involved women spending more time at the test centre [46].

### 3.13. The Impact of Changing Diagnostic Criteria on Obstetric and Neonatal Outcomes 

With the steadily increasing prevalence of GDM, and the serious nature of obstetrical and neonatal outcomes associated with its condition, the burden of these high-risk pregnancies continue to rise. Although we have come a long way towards improving the delivery of GDM care for women with diabetes mellitus in pregnancy, the role of screening and diagnostic criteria continues to remain controversial to date. Over the years, the diagnosis of GDM has evolved from criteria initially developed to predict future maternal diabetes mellitus to recent criteria centred on adverse neonatal outcomes. Evidence from the 2008 HAPO study [8] has demonstrated that the incidence of adverse outcomes occurs on a continuum, as oppose to a definitive inflection point. This has led to great controversy and lack of international unity on setting one global, standard diagnostic threshold for GDM. Although adverse neonatal outcomes are the basis of the IADPSG’s one-step approach, there remains a lack of randomized clinical trials that demonstrate that its application leads to improvements in neonatal outcomes relative to the two-step approach.

Fuelling the controversy, several studies have compared these adverse pregnancy outcomes across the two approaches with divergent findings [44,46,47,48,49,50]. In a retrospective cohort study conducted by Pouliot et al. [44], the investigators compared pregnant women who were screened for GDM using the two-step approach (pre-IADPSG group) to women who were screened using the one-step approach (post-IADPSG group). The authors found that women in the post-IADPSG group were observed to have lower rates of labour induction, pre-eclampsia and offspring admission to the neonatal intensive care unit and concluded that one-step testing was associated with improved pregnancy outcomes. Similarly, in another retrospective cohort study conducted by Sacks et al. [48], the authors compared pregnancy outcomes among women without GDM during pregnancy, untreated women who only met the criteria for the IADPSG’s one-step approach and women who met DC’s criteria for their preferred two-step approach. Women with more severe GDM (higher glucose levels) were treated and excluded in this study. Relative to those without GDM, untreated women who were diagnosed with the two-step approach demonstrated significant increased risk of shoulder dystocia, pre-eclampsia, pre-term births, delivering large-for-gestational age offspring, and delivering offspring with hypoglycemia. Compared to women without GDM, untreated women diagnosed with the one-step approach only demonstrated increased risk of delivering large-for-gestational offspring but none of the other obstetric and neonatal outcomes.

In contrast, Meltzer et al. demonstrated, in a clinical trial of 5142 Canadian women (total sample size) randomized to a GDM screening approach (described earlier), that higher rates of pre-eclampsia (Group 1, 3.5%; Group 2, 3.3%; Group 3, 5.4%; *p* < 0.05) and neonatal hypoglycemia (Group 1, 3.5%; Group 2, 4.2%; Group 3, 6.5%; *p* < 0.05) were observed among women in Group 3 undergoing one-step testing (applying 2013/2018 DC threshold values), relative to those in Groups 1 and 2 that underwent two-step testing for GDM [46,49]. Maternal data were obtained from the McGill Obstetric and Neonatal Database. Furthermore, in a recent population-based cross-sectional study conducted by Shah and Sharifi [47], the authors assessed 90,140 pregnant women in Ontario who underwent a 75 g OGTT between 2007 and 2015. Women were classified as those who met the 2013 DC criteria for the two-step approach and were treated, those who were untreated but would have only met the IADPSG criteria for the one-step approach (but not the two-step thresholds), and those who did not meet the criteria for GDM. Women diagnosed with the two-step approach demonstrated a significant increase in the risk of pre-term births (RR = 1.25, 95% CI 1.15–1.36), primary caesarean section (RR = 1.07, 95% CI 1.03–1.12), and neonatal intensive care unit admissions (RR = 1.21, 95% CI 1.14–1.28) relative to those who would have been diagnosed with GDM using the one-step approach. In contrast, rates of large-for-gestational-age offspring (RR = 0.87, 95% CI 0.82–0.91) and shoulder dystocia (RR-0.80, 95% CI 0.71–0.90) were lower in women who were diagnosed using the two-step approach relative to the one-step approach. In summary, the absence of robust evidence on GDM diagnostic thresholds and their associated short-term and long-term implications on maternal and neonatal outcomes continues to exist to date. Future research should continue to aim towards comparing these serious perinatal outcomes across women undergoing different screening approaches.

### 3.14. Changes to Screening and Diagnosing GDM in the Context of the Coronavirus Disease (COVID-19)

In the context of the current COVID-19 pandemic, anecdotal evidence indicates that both pregnant women and clinicians are increasingly unwilling to undergo or recommend the OGTT as the primary diagnostic tool for GDM [51]. These concerns are based on issues regarding the time spent exposed when visiting clinics (up to two hours), potential need for multiple visits and time spent travelling. Furthermore, a diagnosis for GDM typically warrants the utilization of additional health care visits including diabetes mellitus education, sonogram imaging and routine glucose monitoring, all of which pose additional exposure risk for COVID-19. In response to these valid concerns, a joint consensus statement was released by DC and the SOGC [52], temporarily recommending that Canadian physicians: (a) continue to perform standard GDM screening if there are only minimal disruptions to lab testing and treatment capacity; (b) perform alternative GDM screening using HbA1C > 5.7% and random plasma glucose levels (RPG) > 11.1 mmol/L to warrant a diagnosis of GDM if the pandemic has caused severe disruptions. 

These recommendations are temporary, given the unprecedented burden that the pandemic has inflicted on Canada’s health care system as professional societies work towards producing comprehensive, patient-oriented and safety-motivated criteria. Generally, the revised Canadian recommendations prioritize specificity over sensitivity due to the shift of health care resources towards combatting COVID-19. These criteria are likely to underdiagnose women with GDM and detect only women with markedly elevated levels of plasma glucose [51]. While HbA1c testing poses the advantage of testing mean glucose levels over time and not requiring women to undergo fasting, several critical drawbacks limit its use as the standard of detection. The first main drawback is that HbA1c is less strongly associated with adverse maternal outcomes than mean OGTT glucose levels as demonstrated in the HAPO study. Secondly, the HbA1c test has reduced sensitivity, given that the proposed HbA1c > 5.7% approximates the 99th percentile of the HAPO cohort [8]. Testing using this approach alone would theoretically reduce the incidence of GDM in the HAPO cohort from 17.8% using the DC-recommended one-step approach to approximately 1% [51]. Controversy surrounding the need to reduce RPG diagnostic thresholds also exists among some Canadian physicians. This stems from HAPO study investigators choosing to unblind pre-diabetic women with a baseline RPG > 8.9 mmol/L as a safety precaution [8].

In terms of screening for overt diabetes mellitus, HbA1c and FPG are the standard screening tests implemented during the early stages of pregnancy (prior to 24 weeks). During the COVID-19 pandemic, Canadian guidelines have recommended that these tests remain unchanged for women with multiple clinical risk factors [52]. In addition, routine post-partum clinic follow ups are deferred until after the pandemic with antenatal care recommended to be administered via telemedicine approaches. Perhaps administration of these alternative testing approaches during these times will provide policy makers additional knowledge and experience that may influence and/or re-establish national guideline recommendations in later years.

### 3.15. Voluntary Online Survey Responses

While many of the guideline recommendation disparities over the last thirty years between the national endocrine and obstetrical organizations of Canada have been resolved, individual variation among Canadian physicians may still exist. Given the absence of trials on the effectiveness of improving fetal-maternal outcomes using the proposed thresholds, physicians across Canada may still choose to base their diagnosis of GDM on different criteria (i.e., clinical expertise and opinion) which are typically subjective. 

Overall, the survey was distributed to 105 physicians from the CanDIPS study group and elicited 13 responses (12.4% response rate). Respondents included a diverse pool of physicians in clinical practices across Canada (Montreal, Toronto, London, Saskatchewan and the North West territories), representing ongoing practices across different regions of Canada. 9 out of 13 respondents (69%) applied the two-step approach in their practice, indicating that more physicians were applying the preferred approach as endorsed by recent guidelines. All 9 respondents applied the appropriate 75 g OGTT thresholds recommended by the latest 2018 DC and 2019 SOGC guidelines (Table 1) and only required one abnormal value to diagnose GDM. Furthermore, 8 out of the 9 physicians using the two-step approach applied > 11.1 mmol/L as the criteria for an immediate diagnosis of GDM following a 50 g GCT; one respondent indicated the use of a lower threshold (10.3 mmol/L) as suggested by the earlier 1993/2003 DC CPGs. Four respondents (31%) indicated the use of one-step approach with values corresponding to the thresholds proposed in both the revised 2013/2018 DC and 2016/2019 SOGC guidelines. 

A total of 9 out of 13 of physicians (70%) responded to the survey’s question probing the use of early screening among pregnant women with multiple clinical risk factors. A total of 8 out of the 9 respondents who provided a response (89%) indicated early screening for overt diabetes mellitus prior to 24 weeks of gestation. Among these 8 respondents, 3 indicated the sole use of the typical criteria they practiced to screen at 24–28 weeks’ gestation (one-step or two-step approach), 2 screened only using FPG values ranging between 5.1 and 6.9 mmol/L corresponding to GDM and 3 screened using A1C > 6.5% and FPG > 7.0 mmol/L to detect overt diabetes mellitus.

With regards to changes in clinical practice as a result of the COVID-19 pandemic, 4 out of 13 (31%) physicians provided a response during completion of the survey. One respondent had indicated no changes to their standard practice, two respondents indicated the use of A1C > 5.7 or random plasma glucose (RPG) > 11.1, and one respondent indicated the use of either A1C > 5.7, RPG > 11.1 or FPG > 5.3 to diagnose GDM. Although survey responses when queried on the “changes to screening and diagnosis of GDM during COVID-19” were low (*n* = 4), 75% of physician responses indicated change to their standard practice, consistent with recommendations advised from urgent update statement released by the CPG Steering Committees from DC and the SOGC [52].

Limitations of our survey include a low response rate and the potential for selection bias to influence the distribution of responses. The survey’s contents were distributed solely to voluntary CanDIPS members (a subgroup task force of DC) due to accessibility (RB). Response rates and external validity may be improved in the future by distributing the surveys contents to health care providers that are members of the SOGC or other large-scale, family physician organizations. 

Overall, our survey demonstrated consistent results among a voluntary pool of CanDIPS members in clinical practices across Canada. Responses demonstrated widespread application of the latest national CPGs across our sample and it is possible that the disparities present over the last thirty years may be minimized in current, Canadian clinical practice. 

## 4. Discussion

Historically, there has been debate concerning the optimal approach to screening GDM across Canada. As part of an overview of developments in GDM in the Canadian landscape, this scoping review highlights the history and evolution of national CPGs over the last three decades. We have reviewed the national CPGs published by the SOGC and DC, along with the ideological similarities and differences across each of their updated renditions. We have also reviewed the reported prevalence of GDM and attempted to capture the degree of variability in screening practices among physicians situated across Canada. Both the SOGC and DC have continuously updated their criteria over time, with most physicians in Canada now adopting the latest, nationwide GDM recommendations in their clinical practice. 

With revisions to the latest national CPGs now recommending the use of the one-step approach and only requiring one OGTT abnormal value to conclude a diagnosis, Canada has observed a rise in the estimated prevalence of GDM, as shown in Table 1. Disparities continue to exist, given that Canadian guidelines recommend two different screening approaches (one step vs. two step) for identifying GDM; the lack of consensus contributes to differing estimates of GDM prevalence in Canada. Though SOGC and DC recommendations are frequently guided by expert opinion and consensus, a number of key recommendations are now based on more recent large-scale, prospective cohort studies, such as the 2008 HAPO study [8]. With research aimed at highlighting the reason and level of disparity in Canada over the years, a step forward can be made towards reaching a consensus on a single, unified diagnostic approach to be recommended in future national guidelines.

## Figures and Tables

**Figure 1 ijerph-18-01454-f001:**
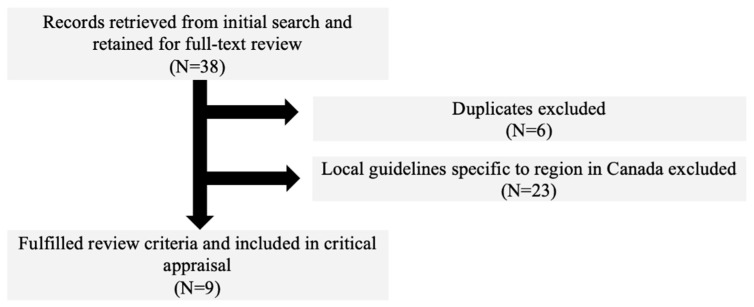
Flow diagram of selection strategy and article reviews.

**Table 1 ijerph-18-01454-t001:** Screening and diagnostic criteria for gestational diabetes mellitus (GDM).

Professional Society, Year	Screening Population	Test	# of Abnormal Diagnostic Values	Fasting Glucose (mmol/L)	1 h Post Glucose Loading (mmol/L)	2 h Post Glucose Loading (mmol/L)	3 h Post Glucose Loading (mmol/L)	Estimated Prevalence of GDM in Canada ^§^
**Society of Obstetricians and Gynaecologists of Canada (SOGC)**								
SOGC, 1992	Universal	2 step 3 h 100g *	2	5.3 or 5.8	10.0 or 10.6	8.6 or 9.2	7.8 or 8.0	3.8–6.5%
SOGC, 2002	Selective	2 step 2 h 75 g	2	5.3	10.0	8.6	--	3.8–6.5%
		2 step 3 h 100 g *	2	5.3 or 5.8	10.0 or 10.6	8.6 or 9.2	7.8 or 8.0	3.8–6.5%
SOGC, 2016	Universal	2 step 2 h 75 g ^†^	1	5.3	10.6	9.0	--	7.0%
		1 step 2 h 75 g	1	5.1	10.0	8.5	--	16.1%
SOGC, 2019	Universal	2 step 2 h 75 g ^†^	1	5.3	10.6	9.0	--	7.0%
		1 step 2 h 75 g	1	5.1	10.0	8.5	--	16.1%
**Diabetes Canada (DC) ^‡^**								
DC, 1998	Selective	2 step 2 h 75 g ^†^	2	5.3	10.6	8.9	--	2.0–4.0%
		2 step 3 h 100 g	2	5.3	10.0	8.6	7.8	2.0–4.0%
DC, 2003	Universal	2 step 2 h 75 g	2	5.3	10.6	8.9	--	3.7%
DC, 2008	Universal	2 step 2 h 75 g	2	5.3	10.6	8.9	--	3.7%
DC, 2013	Universal	2 step 2 h 75 g ^†^	1	5.3	10.6	9.0	--	7.0%
		1 step 2 h 75 g	1	5.1	10.0	8.5	--	16.1%
DC, 2018	Universal	2 step 2 h 75 g ^†^	1	5.3	10.6	9.0	--	7.0%
		1 step 2 h 75 g	1	5.1	10.0	8.5	--	16.1%

* Includes both Carpenter–Coustan and National Diabetes Data Group (NDDG) criteria. The Carpenter–Coustan criteria are the lower, more inclusive thresholds illustrated in this row. ^†^ Preferred approach. ^‡^ Formerly known as the Canadian Diabetes Association. ^§^ Estimates of GDM prevalence as reported in each CPG; derived from observational cohort studies described in the respective guidelines.

## Data Availability

Not applicable.

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
