# Peer review of "Trends in National Canadian Guideline Recommendations for the Screening and Diagnosis of Gestational Diabetes Mellitus over the Years: A Scoping Review"

_ijerph, 2021, doi:10.3390/ijerph18041454_

Round 1
Reviewer 1 Report
A comprehensive summary of the history of approaches to GDM testing and diagnosis in Canada. While a difficult topic to outline succinctly, I feel the authors have produced a very good article showing the complexity of the evidence/guidelines and the variability in approach it has created particularly well.
Some suggestions for review:
1. In abstract - regarding - 'c) estimates of GDM prevalence in Canada over the years' and later discussion.
- I see that there is some comment on prevalence in Table 1 with reference to this in the body of the text. However, given it is one of the targets for discussion, could the authors consider dedicating a paragraph to this as it has for the other points outlined in the abstract. Could the authors consider making greater comment on the jump in prevalence of GDM with the 1 step 75g OGTT approach and impact on resources. In addition, are there any Canadian information for obstetric/neonatal outcomes after the introduction of this approach?
2. p2. Line: 'In the longer term, GDM is also associated with offspring complications such as childhood obesity, dyslipidemia, and offspring diabetes mellitus later in life.11−13'
- Suggest change second offspring to future or adult
3. p2/3. Comprehensive and appropriate search for literature noted.
4. p4. Line: 'Elevated glucose values were selected by O’Sullivan and Mahan since values in this range represent 95% of the distribution (normality) in the pregnant cohort and applying only one SD 25 and corresponding glucose thresholds would result in greater prevalence of GDM which O’Sullivan believed would lead to a rise in psychologic ill-effects (i.e. depression, anxiety, eating disorders) and long-term follow-up of patients with only mild glucose intolerance.7'
- please review sentence and potentially break up to make for easier reading. I.e. along the lines of: The glucose values within two standard deviations were selected by O’Sullivan and Mahan as normal since values in this range represent 95% of the distribution in the pregnant cohort. Applying only one SD 25 and corresponding glucose thresholds would result in greater prevalence of GDM which O’Sullivan believed would lead to a rise in psychologic ill-effects (i.e. depression, anxiety, eating disorders) and long-term follow-up of patients with only mild glucose intolerance.7'
- Mahan is only introduced fully (as Dr Claire Mahan, statistician) in the paragraph below - consider change to introduction in this paragraph.
5. p5. Line 'In 1982, Carpenter and Coustan (CC) proposed replacing the Nelson-Somogyi method used with more accurate enzyme-based assays.31'
- consider full introduction of Drs Carpenter and Coustan (name, role) in keeping with other physicians noted in article
- Also note variably references as Carpenter-Coustan and CC in article. Consider standardising.
6. p5/throughout - note use of CC to represent above but full name for O'Sullivan - consider using Carpenter-Coustan to standardise.
7. p7. Table 1
- nice summary table - makes the many different approaches easier to understand at a glance
- could the authors either a) make a more obvious distinction between the upper SOCG guidelines and lower DC guidelines (so it doesn't read like a progression in time from top to bottom) or b) mix the SOGC/DC guidelines based on year of introduction. I think the former might be better.
8. p8. Line: 'c) the requirement of plasma glucose levels >10.3mmol/L (at 1-hour post-ingestion) following a 50g GCT to warrant an immediate diagnosis of overt diabetes mellitus; and c) two abnormal OGTT values to conclude a diagnosis.'
- please change second c) to d)
9. p10. Consider breaking a) and b) into separate paragraphs for easier reading. a) is a long discussion and the transition into b) is not obvious as the alternative approach
10. p10. Line: 'The IADPSG has endorsed one-step testing as the only approach to diagnosing GDM and have concerns that many women are unable to return following a 50g GCT.'
- this is a repeat from the above paragraph. While I note this comment is followed by Canadian data, could the authors review it ?
11. p10. Line: 'Recommendations from earlier versions of the Diabetes Canada guidelines (1998/2003/2008) also suggested that plasma glucose levels >10.3mmol/L following a 50-g GCT were sufficient to conclude a diagnosis of GDM.
- could authors consider making this a new paragraph
12. p11. Line 'In the context of the current COVID-19 pandemic, anecdotal evidence indicates that both pregnant women and clinicians remain increasingly unwilling to undergo or recommend the OGTT as the primary diagnostic tool for GDM.42'
- consider changing word remain (? to are) as remain and increasingly seem counter-intuitive
13. p11. Line 'These criteria are likely to underdiagnose women with GDM detecting women with markedly elevated levels of plasma glucose.42'
- please review. Possible change - These criteria are likely to underdiagnose women with GDM and detect only women with markedly elevated levels of plasma glucose.42
14. p11. Line: 'Controversy surrounding the need to reduce RPG diagnostic thresholds also exist among some Canadian physicians; these stem from HAPO study investigators choosing to unblind pre-diabetic women with a baseline RPG >8.9mmol/L as a safety precaution
- please consider change from ; to .
15. Survey noted.
- Could the authors provide a denominator of how many members CanDIPs has (ie how many received the online survey) to determine how representative a sample of 13 is? It seems like a very small number of clinicians upon which to base a full discussion, given this is a subgroup of a national body. It is even harder to make interpretable conclusions from the subgroups of this 13 (i.e. 4 survey results etc)
- Also, in Canada, would most of GDM testing be done by a family physician or obstetrician (who may not be a CanDIPS member)? Ie how representative would a survey of CanDIPs members be of what occurs in general practice? I feel CanDIPS members will show those who are more likely to be up to date with evidence and guidelines. Thus the possible recent convergence of practice may not be generalisable.
- I realise that the authors have acknowledged some of the above limitations
Reviewer 2 Report
It was interesting study in which authors reviewed the development of Canadian guideline recommendations for screening and diagnosing gestational diabetes mellitus over the years. The paper was well-prepared. Some minor issues should be addressed further.
- Authors should distinguish difference between screening and diagnosing. I mean they have different implication on pregnant women health. Screening might take effect when finding potential patients and it has implication of public health.
- Voluntary on-line survey was a smaller scale, just 13 people were involved. Why? Authors consider power of this survey? Maybe authors say something on this issue in the methods and discussion.
Reviewer 3 Report
Authors of the article conducted a literature review on Canadian clinical practice guidelines of gestational diabetes mellitus (GDM). The manuscript is well written and after a few minor changes I recommend publishing it in IJERPH.
- Aim is mentioned two times in the introduction (first and last paragraph). Reviewer suggests keeping only the longer one at the end of introduction.
- Introduction, last paragraph: abbreviation of CPG has been resolved in the first paragraph.
- Page 4: “hour after ingestion of the 50 g glucose load was” remove space between “50 g”.
- Page 5: “but minimal false negative results .12 Since the” remove space between “results .12”.
- Page 5: “In fact, 84% of Canadian physicians (84%) had”, please correct
- Page 8: “an OGTT; c) the requirement of plasma glucose levels >10.3mmol/L (at 1-hour post-ingestion) following a 50g GCT to warrant an immediate diagnosis of overt diabetes mellitus; and c) two abnormal OGTT” second one should be a “d” instead of a “c” again.
- Page 9: “incidence in the HAPO cohort.3 9” please check and correct citation (missing comma?)
- Discussion: after the first paragraph the extra break should be removed.
Author Response
Point 1: Aim is mentioned two times in the introduction (first and last paragraph). Reviewer suggests keeping only the longer one at the end of introduction.
Point 2: Introduction, last paragraph: abbreviation of CPG has been resolved in the first paragraph.
Response 1/2: We thank the reviewer for catching this repetition and typographical error. We have deleted the first paragraph as suggested- see tracked changes- and resolved concerns with the CPG abbreviation.
Points 4-9:
Page 4: “hour after ingestion of the 50 g glucose load was” remove space between “50 g”.
Page 5: “but minimal false negative results .12 Since the” remove space between “results .12”.
Page 5: “In fact, 84% of Canadian physicians (84%) had”, please correct
Page 8: “an OGTT; c) the requirement of plasma glucose levels >10.3mmol/L (at 1-hour post-ingestion) following a 50g GCT to warrant an immediate diagnosis of overt diabetes mellitus; and c) two abnormal OGTT” second one should be a “d” instead of a “c” again.
Page 9: “incidence in the HAPO cohort.3 9” please check and correct citation (missing comma?)
Discussion: after the first paragraph the extra break should be removed.
Responses 4-9: We thank the reviewer for highlighting these typographical errors. We have corrected all suggested errors.
Reviewer 4 Report
This is an interesting historical review of Canadian guidelines with respect to gestational diabetes. It includes a small survey of Canadian physicians from the Canadian Diabetes in Pregnancy Study group (13) to ascertain current practices. Also included are current recommendations in the light of the Covid-19 pandemic. Unsurprisingly the authors note a very wide variation in estimated prevalence of GDM from 3.8% to 16.1% depending on the criteria used.
The paper is well written. Reference to the need for adequate carbohydrate intake (at least 150G) for 3 days prior to the OGTT might have been mentioned. It might be the case that some women when told they had an abnormal 50G challenge, modify their diet in concern for their pregnancy outcome. They might then have an inaccurate 75G challenge. Similarly in the one step OGTT, some women may not have had sufficient carbohydrate in the days before.
The discussion might have included some mention of international comparisons. How widespread is the adoption of the IADPSG criteria elsewhere, in Europe, for example.
Author Response
Please see the attachment. I am uploading our response to reviewer 3 here, as well, as I accidentally uploaded my response to reviewer 4 in the section for reviewer 3.

Round 2
Reviewer 2 Report
Authors have addrssed my comments, and the manuscript has been improve much. I have no other concrens.